# EM connectomics reveals axonal target variation in a sequence-generating network

Jörgen Kornfeld[1‡], Sam E Benezra[2,3‡], Rajeevan T Narayanan[4,5,6], Fabian Svara[1], Robert Egger[2,3], Marcel Oberlaender[4,5,6], Winfried Denk[1], Michael A Long[2,3*]

[1]Max Planck Institute of Neurobiology, Martinsried, Germany; [2]NYU Neuroscience Institute and Department of Otolaryngology, New York University Langone Medical Center, New York, United States; [3]Center for Neural Science, New York University, New York, United States; [4]Computational Neuroanatomy Group, Max Planck Institute for Biological Cybernetics, Tübingen, Germany; [5]Bernstein Center for Computational Neuroscience, Tübingen, Germany; [6]Center of Advanced European Studies and Research, Bonn, Germany

*For correspondence: mlong@med.nyu.edu

‡These authors contributed equally to this work

Competing interests: The authors declare that no competing interests exist.

**Abstract** The sequential activation of neurons has been observed in various areas of the brain, but in no case is the underlying network structure well understood. Here we examined the circuit anatomy of zebra finch HVC, a cortical region that generates sequences underlying the temporal progression of the song. We combined serial block-face electron microscopy with light microscopy to determine the cell types targeted by HVC$_{(RA)}$ neurons, which control song timing. Close to their soma, axons almost exclusively targeted inhibitory interneurons, consistent with what had been found with electrical recordings from pairs of cells. Conversely, far from the soma the targets were mostly other excitatory neurons, about half of these being other HVC$_{(RA)}$ cells. Both observations are consistent with the notion that the neural sequences that pace the song are generated by global synaptic chains in HVC embedded within local inhibitory networks.

## Introduction

Neural sequences are central to many models of circuit function (*Diesmann et al., 1999*; *Jin et al., 2007*; *Gibb et al., 2009*; *Fiete et al., 2010*; *Mostafa and Indiveri, 2014*; *Cannon et al., 2015a*; *Rajan et al., 2016*), and neurons often fire sequentially during specific behaviors (*Hahnloser et al., 2002*; *Peters et al., 2014*; *Mello et al., 2015*) or cognitive states (*Pastalkova et al., 2008*; *Harvey et al., 2012*), but the network properties that underlie such dynamics are poorly understood. Here we explore the synaptic connections within the zebra finch HVC, which is central to generating the neuronal activity necessary to coordinate activation of vocal muscles during the highly reproducible courtship song (*Nottebohm et al., 1976*; *Vu et al., 1994*; *Aronov et al., 2008*; *Long and Fee, 2008*). Song progression is paced by HVC$_{(RA)}$ neurons, which project to the primary downstream target area, known as the robust nucleus of the arcopallium (RA) (*Figure 1a*). During the song, an HVC$_{(RA)}$ neuron is either silent or active in the form of a burst of action potentials that occurs at a single precise and cell-specific time (*Hahnloser et al., 2002*; *Kozhevnikov and Fee, 2007*; *Long et al., 2010*; *Vallentin and Long, 2015*). At any moment, it is estimated that about 200 of these 'pacer' neurons are active and can drive the appropriate motor activity (*Fee et al., 2004*), presumably through a set of specific synaptic connections in RA (*Fee et al., 2004*; *Markowitz et al., 2015*; *Lynch et al., 2016*; *Picardo et al., 2016*).

**eLife digest** For us to interact with the world around us, our brains must plan and execute our movements and behaviors. For instance, although speaking is often quite effortless, it is also remarkably complex; all of the muscles in our vocal cords have to be activated at just the right moments to create words. It remains poorly understood how exactly the brain generates such precise timing signals that enable these movements.

A specific portion of the songbird brain allows the bird to sing its song, a process that has clear parallels with human speech. Previous work had demonstrated that this region in the bird's brain acts as a 'clock' for singing behavior, with individual brain cells active at just a single moment, or 'tick'. Little consensus had been reached concerning how this might be achieved.

Kornfeld, Benezra et al. have now used new anatomical methods to better understand how the songbird clock works. A technique called 3D electron microscopy allowed the connections between the neurons in the clock brain region to be seen directly. This revealed that these neurons form direct connections with each other, which is consistent with the idea that one 'tick' can lead to the next and so on, like a series of falling dominoes.

Several mysteries remain to be resolved by future research. First, the connections that Kornfeld, Benezra et al. found are between cells that are quite distant from each other. This arrangement is fundamentally different from many other brain areas where neighboring cells are thought to work together. Second, although these key brain cells form appropriate connections to act as a clock, it is still not clear whether and how the network uses these connections during singing.

By resolving these mysteries, we will establish a new framework for understanding how the brain encodes learned motor gestures that may help to spur innovative new approaches for combatting motor-related deficits due to injury or disease.

It has been difficult to discriminate between different models of sequence generation in HVC, in part because of the unknown connectivity within that nucleus. One class of models uses a synaptic (or 'synfire') chain architecture (*Amari, 1972*; *Abeles, 1991*; *Diesmann et al., 1999*), which can deliver highly reliable and precise timing but requires direct connections between the pacer neurons (*Li and Greenside, 2006*; *Jin et al., 2007*; *Long et al., 2010*; *Cannon et al., 2015a*). Such connections are, however, only rarely seen with paired intracellular recordings, which at the same time showed that HVC$_{(RA)}$ neurons are connected with high probability (>0.50) to nearby inhibitory interneurons (*Mooney and Prather, 2005*; *Kosche et al., 2015*). This observation weakened the case for synfire chain-based sequence generation in HVC and sparked the development of alternative hypotheses that do not require direct connections between excitatory cells (*Yildiz and Kiebel, 2011*; *Hamaguchi and Mooney, 2012*; *Amador et al., 2013*; *Goldin et al., 2013*; *Armstrong and Abarbanel, 2016*; *Hamaguchi et al., 2016*; *Rajan et al., 2016*). There are, however, a number of reasons paired recordings may fail to correctly estimate the connection rate between excitatory cells, among them the severing of axons during slice preparation (*Stepanyants et al., 2009*) and an oversampling of closely spaced neurons (*Jiang et al., 2015*). To avoid this bias, we used a structural approach combining anatomical reconstructions of complete cells in light microscopy (LM) with high-throughput serial block-face electron microscopy (SBEM) (*Denk and Horstmann, 2004*; *Seung, 2009*).

## Results

We used both LM and EM, because anatomically, synapses can only be identified unambiguously in EM, but currently the size of the volume that can be studied by EM is limited to several hundred microns in one dimension (*Helmstaedter, 2013*). This size is too small to explore the full extent of HVC connectivity, given that axon collaterals of HVC neurons ramify widely throughout the nucleus (e.g. *Figure 1—figure supplement 2a*), which is roughly 2000 × 500 × 500 μm³ in size (*Nixdorf-Bergweiler and Bischof, 2007*). We therefore used LM to explore the mesoscale structure of the axonal morphology and EM to analyze synaptic connectivity. To identify HVC$_{(RA)}$ cells, we injected markers into RA that are retrogradely transported, fluorescent Tetramethylrhodamine (TMR, also

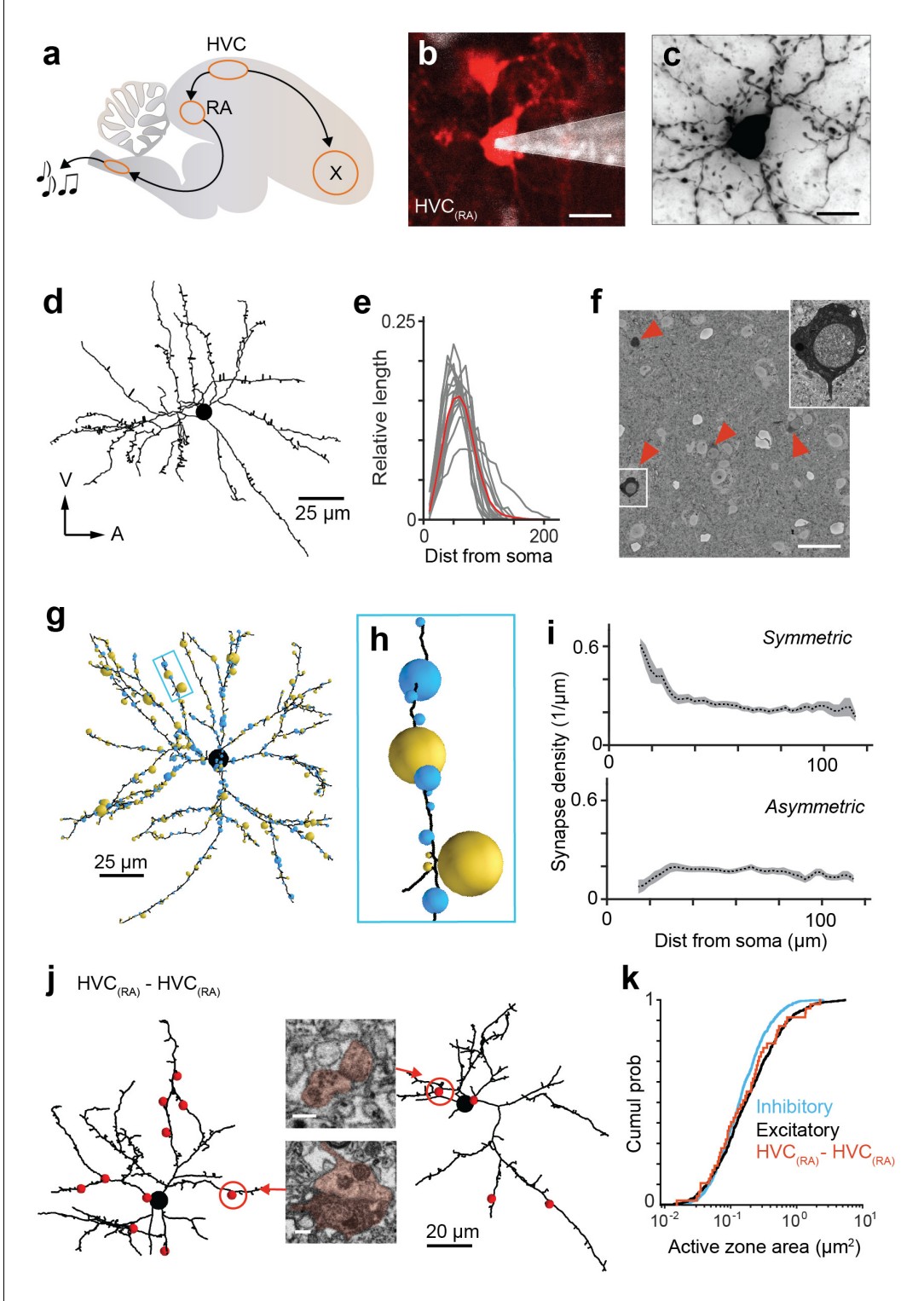

**Figure 1.** Analysis of synaptic inputs onto HVC$_{(RA)}$ dendrites. (a) A schematic of the songbird brain showing HVC and its two main downstream targets, RA and Area X. (b) A backlabeled HVC$_{(RA)}$ neuron (red) during juxtacellular filling (pipette shown in white) guided by 2-photon imaging of fluoro-Ruby. (c,d) A Neurobiotin-filled cell from (b) in brightfield LM after histochemical processing (c) and dendritic reconstruction (d). (e) Normalized count of dendritic path length vs. soma distance for 15 HVC$_{(RA)}$ neurons; individual cells (gray) and average (red). Bin size: 10 μm. (f) Cross-section through a SBEM stack showing BDA-labeled HVC$_{(RA)}$ somata. (g) Inhibitory (blue spheres) and excitatory (gold spheres) synapses onto an HVC$_{(RA)}$ dendrite in the SBEM volume. Sphere cross-sectional areas are proportional to the active zone area. (h) Higher magnification of a dendritic branch from (g). (i) Density of asymmetric and symmetric synapses vs. the distance to the soma. (j) Two HVC$_{(RA)}$ dendrites; red spheres indicate double-labeled synapses, with

*Figure 1 continued on next page*

*Figure 1 continued*

cross sections through two synapses (insets). Inset, cross sections through the synapses circled in red. (k) Active zone size distributions of inhibitory (blue), excitatory (black), and double-labeled (red) synapses. Scale bars are 10 μm in b and c, 25 μm in f, and 0.25 μm in j.

The following figure supplements are available for figure 1:

**Figure supplement 1.** Sample preparation for LM and EM.

**Figure supplement 2.** Synaptic boutons on HVC(RA) axon collaterals.

**Figure supplement 3.** Ultrastructural classification of synapses.

**Figure supplement 4.** The BDA label is inefficient and incomplete.

called fluoro-Ruby) or biotinylated dextran (BDA, *Figure 1—figure supplement 1*), for tissue to be observed in LM or EM, respectively.

To enable the LM-based reconstruction of the entire dendrite and of the axonal collaterals within HVC for single HVC(RA) cells, we used in vivo two-photon microscopy to target (*Komai et al., 2006*) TMR-labeled somata for Neurobiotin labeling (*Figure 1b*). We eliminated all cells (29 of 44) where the labeling intensity varied between different parts of the neurite or where no descending axon could be found. The remaining 15 cells were imaged at $92 \times 92 \times 500$ nm$^3$ voxel size using a transmitted light brightfield microscope (*Oberlaender et al., 2007*) and reconstructed using Neuromorph (see Materials and methods) (*Figure 1c,d*, *Figure 1—figure supplement 1a–e*; *Video 1*). In agreement with other observations (*Dutar et al., 1998*; *Mooney, 2000*; *Kosche et al., 2015*), we found that HVC(RA) dendrites were compact, with $95.0 \pm 2.0\%$ (SEM) of the dendritic path found within 100 μm of the soma (*Figure 1e*). In contrast, the axon collaterals, which were lined with synaptic boutons throughout (*Figure 1—figure supplement 2b*), ramified across HVC. For each cell (n = 15), the dendrite was entirely (100%) confined to HVC, while the axon (with the exception of the branch projecting to RA) was also largely restricted to the boundaries of HVC (97% on average).

To quantify the prevalence of different types of synaptic inputs onto the dendrite of HVC(RA) cells, we next acquired a SBEM data set ($166 \times 166 \times 77$ μm$^3$ overall size, comprising $15104 \times 15104 \times 2661$ voxels, each $11 \times 11 \times 29$ nm$^3$ in size) from the central part of HVC (*Figure 1f*, *Figure 1—figure supplement 1f–j*, *Videos 2* and *3*). All raw data as well as skeletonized reconstructions are available online (*Kornfeld, 2017a*) (https://github.com/jmrk84/HVC_paper; with a copy archived at https://github.com/elifesciences-publications/HVC_paper). Within this volume, 34 somata were positively identified as HVC(RA) neurons by the presence of a BDA-derived electron density (*Figure 1f*). This number is approximately 14% of the expected value of HVC(RA) somata ($240 \pm 28$, SEM), given that there are about $40,000 \pm 3800$ (SEM) HVC(RA) cells (*Wang et al., 2002*) and the total HVC volume is $0.35 \pm 0.024$ mm$^3$ (n = 14, SEM). For 12 of the 34 labeled HVC(RA) neurons, we manually reconstructed (skeletonized) (*Helmstaedter et al., 2011*) the dendrite as far as possible. These reconstructions ranged in dendritic path length from 642 μm to 1956 μm ($1290 \pm 469$ μm, mean ± SD) compared with complete LM-based reconstructions (1438 μm to 4819 μm, mean ± SD: $3187 \pm 997$ μm). Although ~70% (174 out of 248) of dendritic branches reached the boundary of the EM data set and were thus incomplete, 74 branches were completely reconstructed, including their most distal inputs (median ± SD of maximum soma

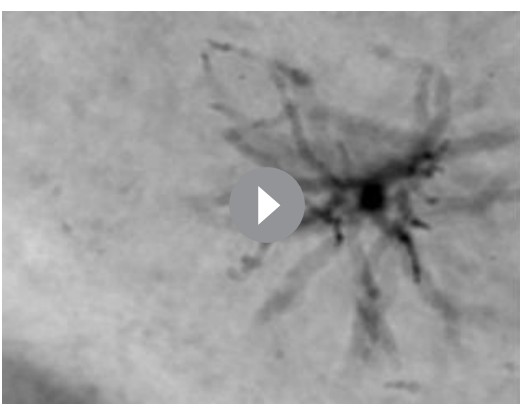

**Video 1.** Video shifting through a z-stack of a sagittal section within HVC, containing a Neurobiotin-filled HVC(RA) neuron stained with DAB. Number of z-sections shown is 144. Voxel size is $92 \times 92 \times 500$ nm.

distances: 90.9 ± 8.6 µm and 116.7 ± 29.4 for EM and LM, respectively). Our reconstructions therefore sample the full gamut of input types. While we do not find any variation of the input type with dendritic distance from the soma beyond a distance of 40 microns (see below), it cannot be completely ruled out that a subtle bias exists that lies below our detection threshold but might be discoverable when using larger data volumes.

We started by classifying for one cell all (1,003) incoming synapses (*Figure 1g–h*) by visually inspecting their ultrastructural details (*Gray, 1959*; *Colonnier, 1968*) (*Videos 4* and *5*). We found that 396 (39.5%) synapses were asymmetric and thus presumably excitatory, and 607 (60.5%) were symmetric (inhibitory) cases. If it was not possible to classify a synapse based on its inspection directly, additional synapses nearby on the same axon were analyzed, since it can be assumed that they are of the same type (*Eccles, 1976*) (*Figure 1—figure supplement 3*). Our synapse classification is reliable: in 19 out of 20 randomly selected test cases, a second expert independently came to the same conclusion and in another set of test cases (8 HVC(RA), 11 HVC(X), and 31 interneuron synapses), where the neuron type was known based on somatic and dendritic morphology (*Figure 2d*, *Figure 2—figure supplement 1*), all synapses were correctly classified by an expert unaware of the cell type.

The dominance of inhibitory synaptic inputs was consistently observed for HVC(RA) cells: when we applied our synapse classification procedure to 97 short dendritic stretches (first and third quartile of stretch length: 13.3 µm and 21.6 µm) randomly selected from eight of the other skeletonized HVC(RA) cells, we found that across neurons the average ratio between excitatory and inhibitory synapses was statistically indistinguishable (p=0.36, one-way ANOVA) from that found in the completely analyzed neuron. Inhibitory synapses were significantly enriched near the soma (68 ± 4% of all synapses at most 40 µm from the soma are inhibitory compared to 57 ± 2%, for synapses beyond that distance, mean ± SEM, p<0.05, Wilcoxon rank-sum test, *Figure 1i*), an observation also made in cortical neurons (*Anderson et al., 1994*). To estimate the number of excitatory and inhibitory synapses that a single HVC(RA) neuron receives on average, we first calculated dendritic synapse densities for all nine analyzed cells separately for asymmetric (0.25 ± 0.02 µm$^{-1}$, mean ± SEM) and for symmetric synapses (0.36 ± 0.02 µm$^{-1}$). To get expected counts per cell, we multiplied these with the full dendritic path length (on average 3.2 mm per neuron), determined from LM reconstructions. Thus, on average well above half of all synapses onto HVC(RA) dendrites are symmetric (59%, 1144 ± 429, mean ± SD) and only 41% are asymmetric (786 ± 311) — a surprising dominance of inhibitory inputs that stands in stark contrast to mammalian cortical neurons (*Beaulieu et al., 1992*; *Peters, 2002*; *Kasthuri et al., 2015*), where the inhibitory synapses are typically found to be at most 20% of the total.

We next inspected all BDA-labeled dendrites emerging from the 12 aforementioned cells for synapses in which the presynaptic axon was labeled, and thus had to come from other HVC(RA) cells (*Figure 1j*). We found 44 such homotypic synapses between HVC(RA) cells (see Materials and methods), but they comprise only about 1% among an estimated total of 3817 ± 926 (SD) incoming excitatory synapses. Their median size (0.21 µm$^2$) and size variation (first and third quartile: 0.10 µm$^2$ and 0.48 µm$^2$), were statistically indistinguishable from those for all asymmetric synapses (0.17 µm$^2$; first and third quartile: 0.08 µm$^2$ and 0.39 µm$^2$, p>0.05, Wilcoxon rank-sum test, *Figure 1k*). One might be tempted to consider the small number of double-labeled synapses as evidence that HVC(RA)-HVC(RA) connections are rare. However, BDA labeled only a small fraction (1/7th) of all HVC(RA) cells in our data set (*Figure 1—figure supplement 4a*) and even for those, axonal collaterals were often incompletely filled (*Figure 1—figure supplement 4b*), suggesting the probability that a given stretch of HVC(RA) axon is labeled could be quite small. To estimate this probability, we created a 300-member set of 1 µm$^3$ cubes

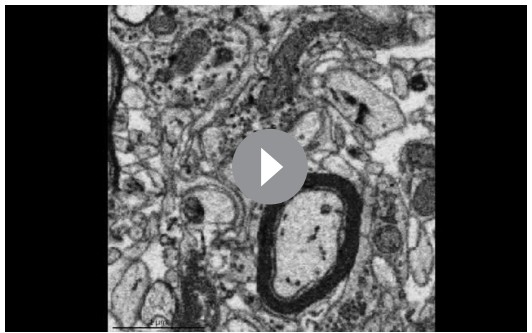

**Video 2.** Video of a subregion of the acquired SBEM dataset, showing the original data resolution (lossy compression). Number of z-sections shown is 100, translating to 2.9 µm.

randomly placed throughout the SBEM volume and measured the total labeled axonal path length they contained. The value obtained (38.6 µm of labeled axon across 300 cubes) is about 13 times smaller than that expected given an estimate of the combined axonal path length (585.6 m) of all 40,000 HVC(RA) cells. The axonal labeling probability of 7.6 ± 1.6% (SEM, see Materials and methods) in turn implies that the homotypic HVC(RA) synapses constitute ~15 ± 4% (SEM) of all excitatory synapses onto HVC(RA) neurons.

We next took a presynaptic perspective to independently estimate the extent of HVC(RA)-HVC(RA) connectivity and used a transsynaptic tracing scheme (*McGuire et al., 1991*) to determine the cell-type of the targets of the outgoing synapses on BDA-labeled axon collaterals (*Figure 2a*). The three main cell types found in HVC (*Dutar et al., 1998*; *Kubota and Taniguchi, 1998*; *Mooney, 2000*) are easily distinguished in LM: Inhibitory interneurons have smooth dendrites with a nearly complete lack of spines (*Mooney, 2000*; *Wild et al., 2005*), and excitatory neurons project to either RA or to the basal ganglia (Area X), with the descending axon clearly recognizable. Even short stretches of dendrite can be reliably ascribed to one of the three types, because the spine density varies widely between but not within them (*Dutar et al., 1998*; *Kubota and Taniguchi, 1998*; *Mooney, 2000*) (*Figure 2b,c*). Dendrites were largely aspinous (0.01 ± 0.01 spines/µm, mean ± SD) for interneurons, densely covered with spines (0.70 ± 0.13 spines/µm) for HVC(X) cells and less so (0.21 ± 0.07 spines/µm) for HVC(RA) neurons. This spine density metric correctly classified 17 out of 18 BDA-labeled HVC(RA) dendrites in EM as well as 11 inhibitory neurons that had been classified using other morphological characteristics (symmetric synapses and a large soma diameter, *Figure 2—figure supplement 1a*). We used this to classify the cell type of postsynaptic dendritic segments (n = 528) transsynaptically traced from nine BDA-labeled axons fully reconstructed in the EM volume. In 41 of 569 cases, the cell type could not be determined. These cases were excluded from further analysis, because the ultrastructure was obstructed by the BDA label (n = 33) or because the recovered dendritic branch was too short (n = 8), see Materials and methods, *Figure 2d*.

When we examined three BDA-stained axons that each emerged from labeled somata in the SBEM dataset (path lengths: 1.37, 0.88, and 0.72 mm), we found that of 121 connections, 115 terminated on dendrites of inhibitory cells but only six onto excitatory cells, four of which being other HVC(RA) cells (e.g., *Figure 2e*). This agrees with the high connectivity found for closely spaced HVC(RA)-interneuron pairs by electrical recordings (*Kosche et al., 2015*) as well as with reports using EM connectomics for other cortical tissue (*Bock et al., 2011*). However, at this density, there would only be about 20 homotypic synapses per HVC(RA) neuron, which is about six times smaller than our estimate derived from the BDA-labeled inputs onto HVC(RA) dendrites.

We then examined BDA-labeled axon fragments that were 'orphaned' (n = 6, path length: 0.56 ± 0.27 mm, mean ± SD), i.e., could not be traced back to their soma and were therefore likely farther away from it. Three of the fragments were synaptically connected to one of the labeled dendrites and four were partially myelinated. We discovered that the prevalence of synapses onto excitatory neurons, and onto other HVC(RA) cells in particular, was much larger for orphaned fragments than for attached axons; increases were 13-fold (HVC(RA)-E), from 5.0% (6 out of 121) to 64.6% (263 out of 407), and 11-fold (HVC(RA)-HVC(RA)), from 3.3% (4 out of 121) to 36.8% (150 out of 407) (*Figure 3a*). HVC(RA) dendrites were often connected by more than one synapse to a labeled axon (17 doubles, 3 triple, and 1 quintuple among 127 analyzed pairs). The much larger (compared to the proximal outputs) fraction of excitatory target cells for the orphans implies that the prevalence of the different target types must depend on the distance from the soma. This would also be consistent with the low connection probability of 0.7% between HVC(RA) cells found in electrophysiological recordings (*Kosche et al., 2015*), where the recorded somata are usually less than 200 µm apart (*Mooney and Prather, 2005*; *Jiang et al., 2015*), while, as our LM reconstructions show, 56 ± 14% (SD) of the axon collaterals' path lies farther than 200 µm from the soma, with some of them ramifying over the extent of HVC (e.g., *Figure 1—figure supplement 2a*).

Can we estimate the distance of an orphan segment to its soma based on local information?

It is apparent from our LM reconstructions that branching becomes less frequent as the distance from the soma increases (*Figure 3b,c*). Consistent with this, HVC(RA) axons in the SBEM data set that were connected to a cell body were much more highly branched (12.4 ± 3.7, mean ± SD, branch points/mm, *Figure 2e*) than most orphaned axon fragments, with an average of only 4.0 ± 4.3 (mean ± SD) branch points/mm. To obtain a quantitative estimate of the distance to the soma and its uncertainty based on the number of branch nodes on a branch and its length we used both a nearest

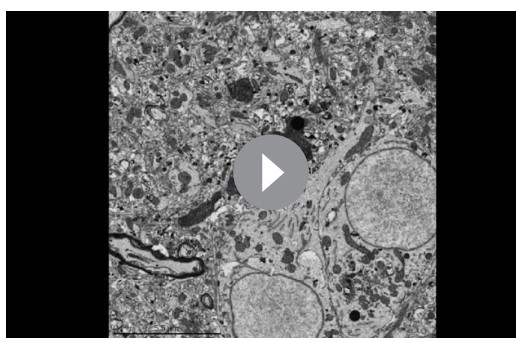

**Video 3.** Video of a subregion of the acquired SBEM dataset, showing a larger field of view with a BDA-labeled HVC(RA) soma (lossy compression). Number of z-sections shown is 200, translating to 5.8 μm.

neighbor (*Figure 3d–g*) and a Bayesian (*Figure 3—figure supplement 1*) analysis (for details see Materials and methods). We found that a synapse was much more likely to be connected to another HVC(RA) cell or to a HVC(X) neuron rather than to an inhibitory neuron if the synapse was farther away from the soma (*Figure 3d*). The transitions between these regimes may well be gradual: One of the orphaned axons (*Figure 3—figure supplement 2*) showed an unusually high branch density (11.4 branch points/mm), suggesting a location close to the soma (16th to 84th percentile: 35.8 to 72.3 μm and also made the majority of its connections (52 of 83, 63%) onto interneurons, twice the fraction seen for the other orphaned axons (32 ± 10%, mean ± SD).

To rule out the possibility that our findings are due to a selection bias, we estimated the fraction of homotypic synapses for the 59 BDA-labeled axon fragments found in the 300-member set (see above), tracing each fragment from the sampling cube until we found two synapses or reached the data set boundary, and determined the postsynaptic cell types. Out of 105 synapses, 65 targeted interneurons, 22 HVC(X) neurons, and 18 other HVC(RA) neurons. Since there are approximately 1111 ± 513 (SD) outgoing synapses inside HVC for each HVC(RA) neuron (given an axon path length of 14.7 mm and a total synapse density of 75.4 synapses/mm), we expect about ~191 ± 88 (SD) homotypic synapses per cell (on average, incoming and outgoing homotypic synapse have to be equal in number), comprising about a quarter (24 ± 4%, SEM) of all incoming excitatory synapses, and nearly half of all outgoing excitatory contacts. The discrepancy between the estimates of the homotypic fraction of incoming excitatory synapses from the dendritic (~15%) and axonal perspective (~24%) might be due to the fact that when counting the number of double labeled synapses, we accepted only those where the labeling of the presynaptic terminal was unambiguous.

How can we be sure that all or at least most of the orphaned fragments belong to HVC(RA) neurons? Since BDA (which is transported in the retrograde direction much more efficiently than anterogradely in all tissues tested, including the zebra finch brain (*Reiner et al., 2000*) was only injected into RA, any labeled axon has to belong to a cell with an axon that connects HVC and RA, as HVC(RA) axons do. If there is indeed a substantial number of cells in RA that project to HVC(*Roberts et al., 2008*), then it is possible that a substantial fraction of the orphaned axons could originate from those cells. To independently confirm the number of RA(HVC) cells, we injected the fluorescent tracer DiI (Invitrogen, Carlsbad, CA) into HVC, which heavily labeled the upstream nuclei NIf and Uva (*Figure 3—figure supplement 3*) but yielded only a small number of labeled somata in RA (125, 163, and 171, respectively, in three birds), approximately one for every 200 HVC(RA) neurons on average. To account for the density of labeled axon in our EM volume, each those cells would need a total axon path of ~4 m in HVC, which appears unlikely given that the extensively ramifying HVC(RA) axons have a length of only ~0.015 m.

## Discussion

We have shown that the synaptic architecture in HVC contains a density of connections between HVC(RA) neurons that might be sufficient to support a synaptic-chain model, whereby precisely timed sequences of action potential bursts in HVC(RA) neurons are generated by a wave of activity propagating via synaptic connections among these neurons without the need for inhibition-mediated propagation of activity (*Yildiz and Kiebel, 2011*) or to involve structures outside HVC (*Hamaguchi and Mooney, 2012*; *Goldin et al., 2013*; *Hamaguchi et al., 2016*).

While we estimate that 25% of excitatory inputs to HVC(RA) neurons are homotypic, the sources of the remaining synapses are unknown. It should be a central goal of future efforts to quantify the relative number of connections from these regions (e.g., Uva, NIf, etc.) at the level of single HVC(RA)

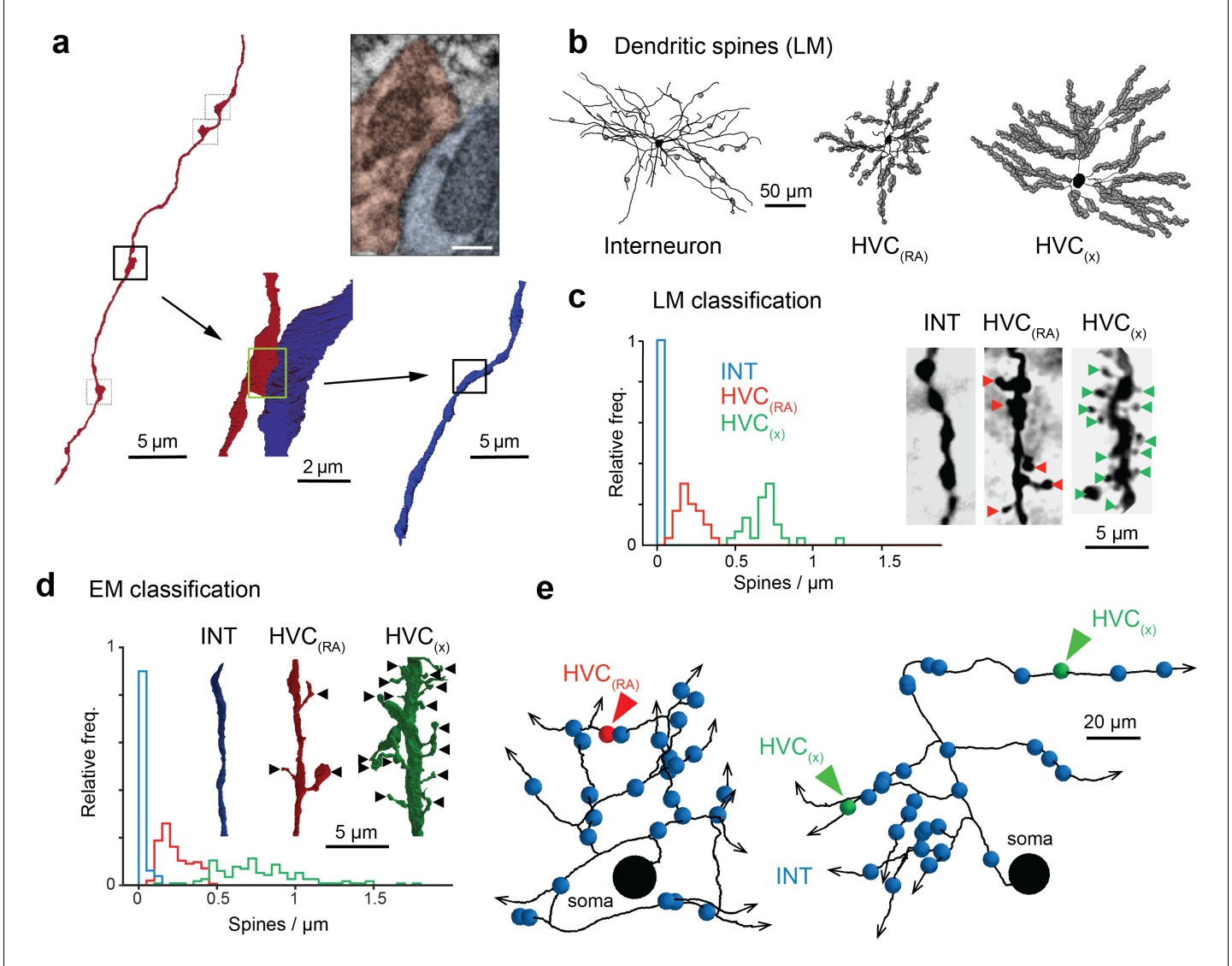

**Figure 2.** Classification of postsynaptic targets. (**a**) A BDA-labeled axon with four synaptic boutons (boxes). One bouton and its postsynaptic structure labeled in red and blue, respectively: In cross section (top right) and as a surface reconstruction (bottom center). (**b**) Dendrites from an inhibitory interneuron, an HVC(RA) neuron, and an HVC(X) neuron (left to right) in LM. Spine locations are indicated by grey spheres. (**c,d**) Spine densities for each of these neuron classes from LM (**c**) and EM (**d**) reconstructions. Insets show examples with spines indicated by arrowheads. (**e**) SBEM-based reconstructions of two HVC(RA) somata with their proximal axons. Blue, green, and red spheres mark the location of synapses with inhibitory interneurons, HVC(RA) neurons, and HVC(X) neurons, respectively. Scale bar is 0.25 μm in a.

The following figure supplement is available for figure 2:

**Figure supplement 1.** Morphological markers of interneurons.

neurons. That said, many of these connections, such as auditory afferents (*Vallentin and Long, 2015*), collaterals from HVC(X) neurons (*Scharff et al., 2000*), and descending fibers from NIf are unlikely to play a role in motor patterning, since removal of NIf does not disrupt the song (*Cardin et al., 2005*). The precise role of Uva, a thalamic region also directly projecting to HVC (*Nottebohm et al., 1982*), remains to be determined (*Coleman and Vu, 2005*; *Hamaguchi et al., 2016*).

Somewhat surprisingly, the low rates of pairwise connectivity seen in electrophysiological recordings (*Mooney and Prather, 2005*; *Kosche et al., 2015*), which previously had been interpreted as

evidence against a direct synaptic chain (*Armstrong and Abarbanel, 2016*), are not inconsistent with our estimate that each HVC$_{(RA)}$ neuron receives a significant amount of its excitatory input from other HVC$_{(RA)}$ neurons. The reason is that with the around 200 homotypic inputs per cell, the probability to be connected to any one of around 40,000 HVC$_{(RA)}$ neurons can be at most 0.5%. The question remains what fraction of those inputs are true 'chain' synapses in that the presynaptic cell's activity immediately precedes that of the postsynaptic cell, but our study demonstrates that the anatomical substrate for the chain model exists.

An important next step will be to combine functional imaging with volume EM to directly test whether an HVC$_{(RA)}$ cell receives more numerous or stronger direct homotypic inputs from cells that fire immediately prior to its own activity. In fact, a recent study describes how calcium activity can be imaged in the singing bird (*Picardo et al., 2016*), a crucial step in that direction. One potential difficulty stems from our finding that HVC$_{(RA)}$ neurons preferentially form distal connections, indicating that the timing circuitry in HVC is distributed and therefore requires a large EM volume (as much as 500 million μm$^3$, compared to 2 million μm$^3$ in our volume) for its complete reconstruction. It might take the better part of a year merely to acquire the raw data (*Schalek et al., 2016*). While even a few years ago it seemed impossible to analyze such an amount of data within a reasonable time frame, recent progress in the automation of segmentation are encouraging (*Berning et al., 2015*; *Januszewski et al., 2016*; *Beier et al., 2017*; *Dorkenwald et al., 2017*).

Our finding that connections near the soma are often onto inhibitory neurons suggests that inhibition plays an important role in sequence generation, which is further supported by the large overall fraction of inhibitory inputs. One function of those inhibitory connections could be to decorrelate excitatory activity in space and time: Not only are nearby HVC$_{(RA)}$ neurons rarely connected and thus unable to drive each other, but even when driven by a common input, only the cell(s) with the strongest input(s) will continue to fire in the face of the winner-take-all effect due to the strong reciprocal inhibition (*Figure 3h*). Winner-take-all behavior is normally associated with certain cognitive tasks (*Hopfield and Tank, 1985*; *Lundqvist et al., 2016*), such as decision making (*Usher and McClelland, 2001*). In HVC, it may help to prevent local clusters of activity, which could lead to leakage across different chains passing through adjacent excitatory neurons. An altogether different role for local inhibition may be the improvement of temporal precision by sharpening burst timing through recurrent inhibition (*Hahnloser et al., 2002*; *Long et al., 2010*; *Cannon et al., 2015a*).

Inhibition may, furthermore, have a central role in shaping the distance dependency of postsynaptic targets during circuit development without the need for molecular cues (*de Wit and Ghosh, 2016*). Instead, the architecture we observed may arise naturally from a pattern that initially follows Peters' rule (*Braitenberg and Schüz, 1998*), which predicts synaptic connections between cell types with intermingled axonal and dendritic arbors (*Rees et al., 2017*), but is then refined as the

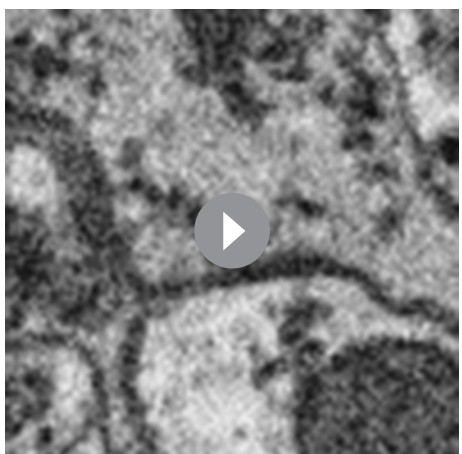

**Video 4.** Video of a z-stack of 18 consecutive images (100 × 100 pixels) showing a symmetric synapse. Voxel dimensions: 11 × 11 × 29.

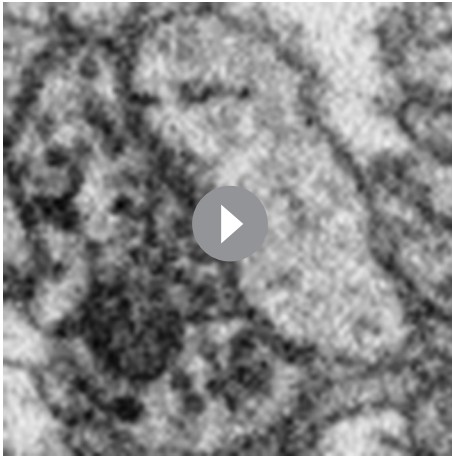

**Video 5.** Video of a z-stack of 18 consecutive images (100 × 100 pixels) showing an asymmetric synapse. Voxel dimensions: 11 × 11 × 29.

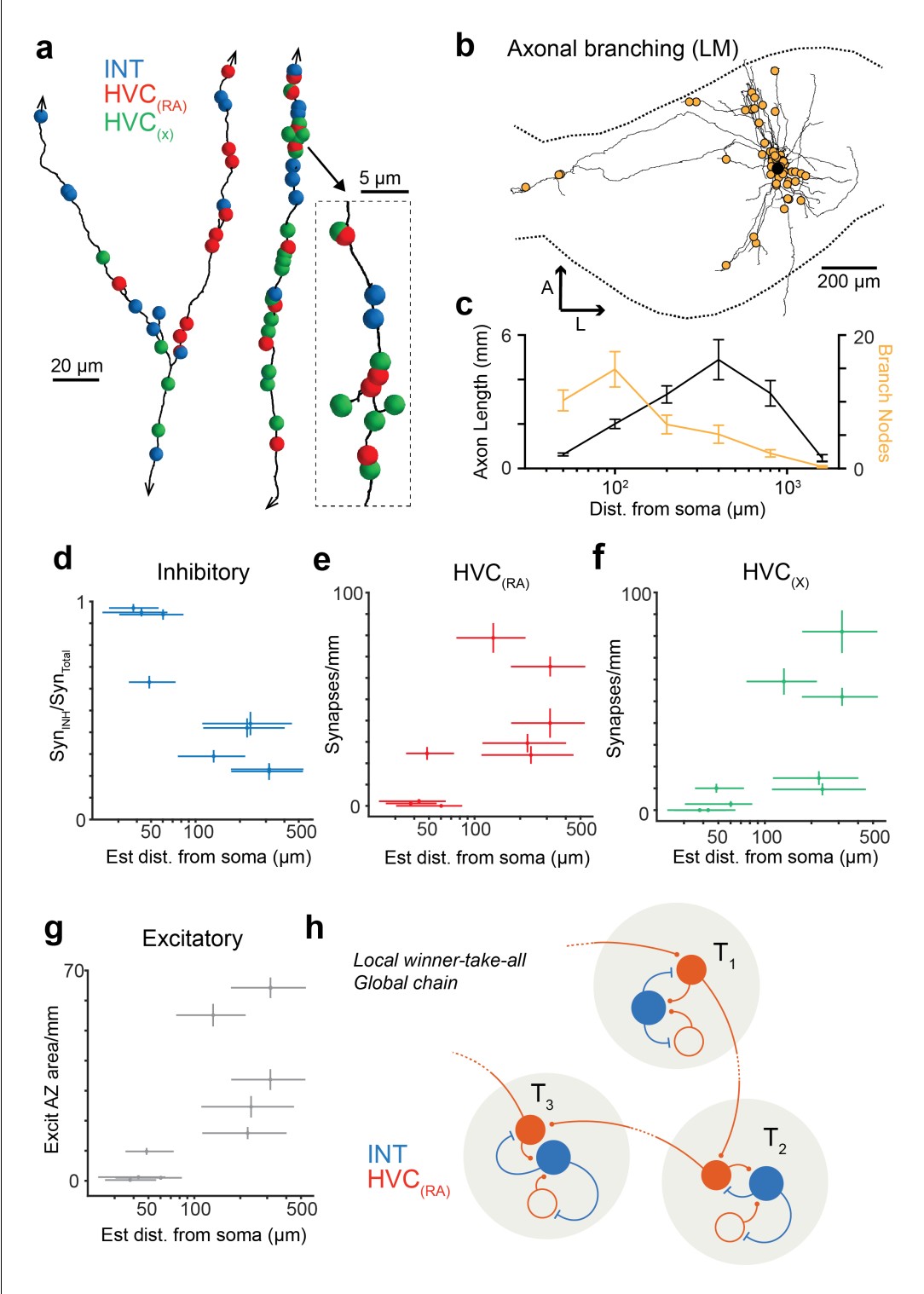

**Figure 3.** Spatial variation of postsynaptic cell type. (**a**) SBEM-based reconstructions and synaptic targets for two orphaned axon segments. Colored spheres mark the locations and types of synapses. (**b**) Axon collaterals (LM-based reconstruction) of an HVC(RA) neuron with branch nodes (gold circles), the soma (black circle), and the HVC border (dashed lines). (**c**) Mean axon length (black) and branch node densities (gold) vs. soma distance (n = 15 cells). (**d**) The ratio of synapses onto inhibitory interneurons vs. estimated distance from the soma (p<0.005, Pearson's correlation). (**e,f**) The density of synapses onto HVC(RA) (**e**) and HVC(X) (**f**) vs. estimated distance from soma (p<0.002, Pearson's correlation, combining HVC(RA) and HVC(X) values). (**g**) Total synaptic size (summated active zone area, $\mu m^2$/mm) onto excitatory neurons vs. estimated distance of the presynaptic axon from the soma (p<0.05, Pearson's correlation). Vertical error bars: SEM of the Poisson-distribution means estimated from the number of synapses on each axon

*Figure 3 continued on next page*

*Figure 3 continued*

segment (**e–g**) or the SEM of an assumed underlying binomial count distribution (**d**). Horizontal error bars from quantiles 0.16 to 0.84 of the distance distribution based on the nearest neighbor sampling approach (see Materials and methods). (**h**) Proposed circuit architecture. HVC$_{(RA)}$ neurons (red) target inhibitory interneurons (blue) proximally and other HVC$_{(RA)}$ neurons distally.

The following figure supplements are available for figure 3:

**Figure supplement 1.** Synaptic properties of HVC$_{(RA)}$ axons, using a Bayesian approach to estimate distance from soma.

**Figure supplement 2.** A SBEM-based reconstruction and synaptic targets for an orphaned axon with high branch density.

**Figure supplement 3.** A small population of RA neurons project to HVC.

interneurons increasingly prevent the co-activation of nearby excitatory cells, thereby destabilizing connections between them while leaving more distant connections intact. Such a preferentially distal connectivity would also favor more widely distributed synaptic chains, which could have the added benefit of relying more on axonal propagation delays for sequence timing (*Budd et al., 2010*). Overall, the observed synaptic architecture shows some resemblance with local inhibitory/excitatory networks linked by long-range excitatory/excitatory connections (coupled winner-take-all modules) that have been shown to make computational models of cortical sequence generation more robust (*Binas et al., 2014*; *Mostafa and Indiveri, 2014*).

## Materials and methods

### Animals

We used adult (>90 days post hatch) male zebra finches that were obtained from an outside breeder and maintained in a temperature- and humidity-controlled environment with a 12/12 hr light/dark schedule. All animal maintenance and experimental procedures were performed according to the guidelines established by the Institutional Animal Care and Use Committee at the New York University Langone Medical Center.

### Surgery

To label only neurons that projected from HVC to the robust nucleus of the arcopallium (RA), we injected lysine-fixable retrograde dextran tracers (Invitrogen) conjugated to either Tetramethylrhodamine (fluoro-Ruby, mol. Weight: 10,000) or biotin (BDA, mol. weight: 3000) for preparations to be inspected with light microscopy (LM) or electron microscopy (EM), respectively. We injected 200 nL of either Fluoro-Ruby (50 mg/mL) or BDA (100 mg/mL) into RA of anesthetized (1–3% isoflurane in oxygen) zebra finches using an injection system (Nanoject, Drummond Scientific, Broomall, PA) outfitted with a glass injection pipette (tip diameter: 30–40 µm). RA was targeted using stereotaxic coordinates (2.30 mm lateral and 1.85 mm posterior from the midsagittal sinus) and success in finding the RA region was confirmed by observing characteristic spontaneous activity (*Long and Fee, 2008*) using a carbon-fiber electrode (Carbostar-1, Kation Scientific, Minneapolis, MN) and an extracellular amplifier (NPI Electronic Instruments, Germany).

For in vivo imaging and dye loading, we first had to enable optical access to HVC. To accomplish this, a craniotomy (1 mm x 1 mm) was prepared over HVC. The underlying dura was then carefully removed with a flame sharpened tungsten wire (starting diameter: 0.5 mm). A small drop of saline buffer was applied to the exposed brain, followed by a 3 mm-diameter round cover glass (#0 thickness, Warner Instruments, Hamden, CT) as an optical window, which was first secured to the surrounding skull by applying light-curable acrylic (Flow-IT ALC; Pentron Clinical Technologies) around the edges of the glass. Dental acrylic (Cooralite Dental MFG, Diamond Springs, CA) and cyanoacrylate were then added to permanently and stably attach the cover glass to the skull. A small metal head plate with two tapped holes was then implanted at the anterior part of the skull using dental acrylic for head fixation.

## 2-Photon guided cell labeling

Juxtacellular labeling (*Pinault, 1996*; *Narayanan et al., 2015*) with Neurobiotin (Vector Labs, Burlingame, CA) was used to fill individual RA-projecting HVC (HVC$_{(RA)}$) neurons out of a population that had been retrogradely labeled from RA with fluoro-Ruby in vivo. After waiting at least 48 hr following the injection of the retrograde tracer into RA, two-photon imaging (*Denk and Webb, 1990*) was used to identify the target cell and guide the pipette. On the day of single-cell labeling, a small pipette access hole (~400–500 μm) was drilled in the glass coverslip immediately lateral to the target recording region using a carbide bur drill bit (1/4 FG-100; Johnson-Promident). Glass pipettes were fabricated using a horizontal puller (P97, Sutter Instrument Company, Novato, CA) and had a final resistance of 4–5 MΩ when loaded with internal solution that consisted of 150 mM K-Gluconate (Sigma-Aldrich, St. Louis, MO) and 3% Neurobiotin. The microscope (MOM, Sutter Instrument Company) was of the moveable objective design (*Euler et al., 2009*) and was controlled using ScanImage (*Pologruto et al., 2003*) 3.8 with a 16x/0.8 NA water immersion objective (Nikon, Japan). Pipettes were made fluorescent either by adding 40 μM of Alexa 488 (Invitrogen) to the internal solution or by coating the pipette with green fluorescent quantum dots (*Andrásfalvy et al., 2014*). The activity of HVC$_{(RA)}$ neurons was recorded (IR-183, Cygnus Technology Inc, Delaware Water Gap, PA), and cells were filled with Neurobiotin by applying 1000–1500 positive current pulses with an amplitude between 3 and 15 nA and a duration of 200 ms delivered at a frequency of 2.5 Hz.

## Histological procedures (LM)

Birds were anesthetized with pentobarbital sodium and perfused transcardially with 4% w/v paraformaldehyde (EMS) at least one hour after dye loading to permit adequate Neurobiotin diffusion. Brains were removed from the skull using a surgical scoop, immersed in 4% paraformaldehyde for 3–5 days to achieve thorough fixation, and incubated in phosphate buffer for an additional 1–3 days to decrease endogenous peroxidase activity. To prepare sections, the brain was cut across the midline, mounted on the sagittal surface with cyanoacrylate, and stabilized with 3% agarose. Parasagittal sections (100 μm thickness) of HVC were cut using a vibratome (Leica VT1000S). Slices were washed five times with phosphate buffer and treated with 3% $H_2O_2$ to further reduce endogenous peroxidase activity. Slices were then immersed overnight at 4°C in a solution containing avidin/biotin complexes and 0.5% Triton X-100 in phosphate buffer (Vector Labs and Sigma-Aldrich, respectively) to tag the Neurobiotin with peroxidase complex. On the following day, slices were washed five times with phosphate buffer and then immersed in a solution containing 2.3 mM diaminobenzidine (DAB, Sigma-Aldrich) and 0.01% $H_2O_2$ in phosphate buffer to label processes containing Neurobiotin. Slices were then washed and mounted on slides with Vectashield (Vector Labs) or Mowiol (Sigma-Aldrich) mounting medium.

To quantify the number of HVC-projecting RA neurons, we injected a retrograde tracer into HVC (DiI, Invitrogen D3911; 46 nL total injection volume) that labels neurons with high efficiency in zebra finches (*Scott et al., 2012*). Following a two-day incubation period, animals were perfused with 4% paraformaldehyde, and 100 μm sagittal sections were cut across the entirety of RA, Nucleus Interfacialis (NIf), and nucleus Uvaeformis (Uva). Sections were mounted on slides using Vectashield (Vector Labs) and imaged with a confocal microscope (LSM 800, Zeiss, Germany; excitation / emission: 551/569 nm) using a 20x objective (0.8 NA). The z-stacks of retrogradely labeled RA$_{(HVC)}$ neurons were captured across the extent of RA, and the position of each cell was manually marked using the landmark function in Amira.

## LM imaging

Only well-filled HVC$_{(RA)}$ neurons were selected for reconstruction, specifically those in which the soma, dendrite, and axon were all labeled (even if faintly) without interruptions and with clearly labeled dendritic spines and presynaptic boutons were selected for high resolution LM imaging with a custom-designed high-resolution mosaic/optical-sectioning brightfield microscope system (*Oberlaender et al., 2007*). In brief, a transmitted light brightfield microscope (Olympus BX51, Olympus, Japan), equipped with a motorized x-y-z stage (Maerzhaeuser, Germany), a narrow bandpass (546 ± 5 nm) illumination filter and a 100x magnification oil-immersion objective (numerical aperture 1.4) was used to acquire image stacks from consecutive 100 μm thick brain sections. For each section, a 3D mosaic of images (e.g., 10 × 15 fields of view) covering the entire HVC was

acquired at 92 × 92 nm pixel size and in steps of 500 nm mechanical defocus. Next we applied a linear image restoration algorithm (Tikhonow-Miller) using the Huygens software package (Scientific Volume Imaging, Netherlands). By inverting the gray values of the brightfield image stacks they could be treated as fluorescent data with an emission wavelength of 546 nm. The deconvolution used a point-spread-function that takes the optical properties of biocytin-labeled brain tissue into account (*Oberlaender et al., 2009*). Deconvolved image stacks were then downsampled by a factor of two in x/y, yielding a final voxel size of 184 × 184 × 500 nm before axonal reconstruction. To quantify the bouton density, subvolumes that contained primarily horizontal (i.e. within the image plane) axonal branches were acquired at 200 nm focus increments and used without deconvolution.

## Neuron reconstructions (LM)

Neuronal branches (dendrites and axons) were reconstructed in 3D using NeuroMorph (*Oberlaender et al., 2007*). Automated tracing results from each histological section were manually proof-edited using FilamentEditor (*Dercksen et al., 2014*), custom-designed based on Amira visualization software (FEI-VisualizationSciencesGroup). In brief, maximum-intensity z-projections of the original image stacks were superimposed onto automatically generated 3D skeleton tracings of all putative neuronal branches contained within the imaged volume and segmented objects that had no correspondence in the projection image were manually deleted (*Dercksen et al., 2014*). Fragmented segments were spliced, and axonal branches were classified as 'dendrite' or 'axon' based on whether, respectively, spines or boutons were visible in the projection images. Whenever a neuronal branch reached one of the borders of the imaged volume, additional image stack regions were acquired that allowed us to follow the branch further. To account for shrinkage during histological processing, the reconstruction was scaled to match the thickness of 100 µm, as defined by the vibratome. The scaled 3D tracings from all consecutive sections were then combined and manually aligned using the FilamentEditor. The z-coordinate of each point was then replaced by the average of nine points (the point itself and the four adjacent points in each direction) and resampled to a point spacing of about 1 µm. Smoothing in z and downsampling make path length measurements comparable to manual tracing results using Neurolucida Software (Microbrightfield, Williston, VT). The NeuroMorph and FilamentEditor tools enable tracings that are independent of the experience of the human operator, with an interuser-variability of approximately 20 µm per 1 mm axonal length (*Dercksen et al., 2014*). The borders of HVC were manually traced in each 100 µm tissue section using Neurolucida.

## Analysis of LM reconstructions

The fraction of dendritic length contained within a certain distance of the soma was determined by conducting a spherical Sholl analysis (*Sholl, 1953*) in Neurolucida (Microbrightfield). The proportion of axonal pathlength both within HVC and within a 200 µm radius from the soma was computed in Amira for each neuron using the ZIB extension package (*Egger et al., 2014*). Axonal boutons and dendritic spines were annotated manually in Amira using high-resolution LM stacks. The location of each bouton or spine was marked in 3D and aligned in Amira to the corresponding branch reconstruction. Spine-densities were calculated for each branch by dividing its total spine count by its path length. Branch nodes (points where the axon bifurcates) were manually located in the reconstructions using Amira. Branch nodes for which one of the daughter branches was <15 µm in length were not included in this analysis.

## Histological procedures (EM)

The bird used for the EM experiments was transcardially perfused in a way that preserves the extracellular space and leads to minimal shrinkage (JK, unpublished observations), by using high pressure and the following fixative solution: 0.07 M sodium cacodylate (Serva, Germany), 140 M sucrose (Sigma-Aldrich), 2 mM $CaCl_2$ (Sigma-Aldrich) with 2% paraformaldehyde and 2% glutaraldehyde (Serva) added (*Cragg, 1980*). The brain was removed and, using a vibratome (Leica VT1000S), cut into slices each about 200 µm thick. One of the slabs that centrally intersected HVC was selected and post-fixed in the same solution overnight, rinsed several times with cacodylate buffer and permeabilized in a 30% sucrose solution by exposing it to one freeze-thaw cycle in liquid nitrogen. Residual peroxidase activity was suppressed by soaking the sample in 3% $H_2O_2$ for 30 min before

labeling the sample with an avidin-peroxidase complex and DAB, as described in a previous section. The sample was then rinsed several times in cacodylate buffer. Heavy metal staining was added through a conventional ROTO protocol using the following steps interspersed with rinses in cacodylate buffer (after first Osmium step) or $H_2O$ (all others): 2% $OsO_4$ (Serva), reduced with 2.5% potassium hexacyanoferrate(II) (Sigma-Aldrich) 2 hr, room temperature; 1% thiocarbohydrazide in $H_2O$, 1 hr, 58°C (Sigma-Aldrich); 2% $OsO_4$, 2 hr; 1.5% uranyl-acetate in $H_2O$, 53°C (Serva); 20 mM lead-aspartate, 2 hr, 53°C (Sigma-Aldrich) (*Seligman et al., 1966*; *Karnovsky, 1971*; *Walton, 1979*). Dehydration was performed using an ethanol series with 10, 15, 10, 10 min at 70%, 100%, 100%, and 100% ethanol (Electron Microscopy Sciences). The sample was infiltrated with epoxy monomer (epon hard, Serva) (*Glauert and Lewis, 2014*) dissolved in propylene oxide (Sigma-Aldrich) for 3 hr and for 3 hr with pure monomer before final embedding and curing (48 hr at 60°C). The sample was then trimmed and glued with epoxy to a custom-made aluminum holder and trimmed into a pyramidal-shape before gold coating for better conductivity.

## SBEM imaging and data preprocessing

We performed serial block-face electron microscopy (*Denk and Horstmann, 2004*) at $11 \times 11 \times 29$ nm voxel size using a scanning electron microscope with a field-emission cathode (UltraPlus, Zeiss, Germany) equipped with a custom-built in-chamber microtome in high-vacuum (raw and effective voxel rates were 5 and 2.1 MHz respectively) at a dose of 10.3 electrons/nm², 2 kV landing energy with a custom back-scatter electron detector and amplification system optimized for fast acquisition speeds. Before each cut, a subregion of the block face was imaged using four overlapping micrographs resulting in an image stack. Images were registered by affine transformations (https://github.com/billkarsh/Alignment_Projects) (*Scheffer et al., 2013*; *Karsh, 2016*) and converted to a KNOSSOS (www.knossostool.org) data set for reconstruction and browsing with custom Python code (https://github.com/knossos-project/knossos_python_tools/tree/master/knossos_cuber) (*Kornfeld, 2017b*). Copies of the software are archived at https://github.com/elifesciences-publications/Alignment_Projects and https://github.com/elifesciences-publications/knossos_utils).

## Neuron reconstructions (EM)

Each annotator received at least 10 hr of training and was considered an expert after one year of annotation experience. BDA-labeled neurons, using the soma as a starting place, were skeletonized within the EM stack in KNOSSOS by an expert annotator, and errors were corrected by the same individual in a second pass, which was also used for synapse annotation. All BDA-labeled axons, including orphaned axons, were traced by at least two independent annotators and discrepancies were resolved by an expert that had not participated in the initial annotation. Synapses on each axon were then labeled (see synapse identification) and proofread by an expert annotator who excluded cases where the BDA-label obscured the ultrastructure. The remaining synapses were used to seed the tracing of the postsynaptic dendrite segment. Annotators were instructed to reconstruct the postsynaptic dendrite to the end of the branch in one direction and to the next main branch point in the other direction. All dendritic-branch tracings were proofread by an expert and only included if at least a minimum path length of 10 μm could be reconstructed. All EM reconstructions were analyzed and visualized with custom Python code using the Mayavi2 (Enthought) library (*Kornfeld, 2017a*, *Kornfeld, 2017b*).

## EM synapse annotation

Synapses were labeled by an expert annotator and classified as symmetric or asymmetric (*Videos 4* and *5*, *Figure 1—figure supplement 3*). Active-zone' diameters 'were quantified by measuring - with KNOSSOS - the cross-sectional length of the synaptic thickening in that plane and principal viewing orientation (x, y, or z) in which the contact cross section appeared largest. Diameters were then converted to areas by assuming a circular synaptic contact.

## Classification of postsynaptic cell type

To estimate dendritic spine density, a stretch of the postsynaptic dendrite (>10 μm) was selected that often included the place where the axon was in contact with the dendrite. We counted as a spine every skeleton branch with a length greater than 1 μm that emerged from the dendritic shaft.

Some postsynaptic protrusions found on interneurons contained multiple synapses (e.g., *Figure 2—figure supplement 1b*). Therefore, spines were defined as receiving no more than one synapse at their ends by three independent annotators. The resulting spine density $D_{spine}$ (in $\mu m^{-1}$) was used to classify the dendritic stretch as belonging to an interneuron ($D_{spine}$ < 0.11), $HVC_{(RA)}$ (0.11 < $D_{spine}$ < 0.46), or $HVC_{(X)}$ neuron (0.46 < $D_{spine}$). To detect dendritic reconstructions that were traced from separate synapses but belonged to the same dendrite, we detected overlap between skeletons using the following criterion: a node was considered to overlap another skeleton if it was less the 400 nm from any edge of all other skeletons. Dendritic reconstructions were defined as belonging to the same neuron when at least 25% of their nodes overlapped. Since the postsynaptic dendritic reconstructions were never complete (i.e. only parts of the entire neuron could be reconstructed), our analysis could only positively identify reconstructions as belonging to the same cell. For dendrites that were found to belong to the same cell (grouped together after being traced from different synapses), spine density was averaged before classification.

## Estimating the axon-to-soma distance

We used two different ways to estimate the distance between an orphaned branch and its soma from its number of branch nodes inside the EM volume, both based on the LM observation that the density of branch nodes, $D_b$, varies with soma distance (r) (*Figure 3c*). The first way used a Bayesian approach to calculate the probability distribution over r, given a branch of length *l* and a branch-node count of *N* (*Figure 3c*), which can be used to estimate, as needed, mean, median, variance or any quantile for r:

$$P(r|N,l) \propto P(N,l|r) * P_a(r),$$

whereby

$$P(N,l|r) = \frac{(D_b(r)*l)^N * e^{-D_b(r)*l}}{N!},$$

which assumes that the branch nodes are placed independently from each other and are, therefore, Poisson distributed with a node-count expectation value of $\lambda = D_b(r)*l$. Fitting the LM measurements to an exponential gave $D_b(r) = \left(35.448 * e^{-\frac{r}{43.5mm}} + 0.613\right) mm$. The Bayesian prior, $P_a(r)$, i.e. the probability that an axon segment is found at a distance between r and r± from its soma, was estimated by applying Gaussian kernel density estimation (Python scipy.stats.gaussian_kde, scott bandwidth selector) to the LM based axon distribution measurements.

The other way to relate r to N and l is to sample the LM data directly: We divided each of the 15 LM stacks into volumes shaped identically to the EM volume and recorded for each volume and for all contained orphaned branches their lengths, distances from the soma, and branch-node counts. Only branches that both entered and left the sampled subvolume were considered (about 95% of the total) because all of the reconstructed orphaned branches in the EM volume also had that property. This was repeated with the origin of the division grid shifted in 10 μm increments along all three axes resulting in 17 × 17 × 8 different divisions for each LM stack. For a given orphaned branch in the EM volume, we selected all those sample branches that had the same node count and a length within ± 10%. The distribution of their soma distances was then used in the same way as the probability distribution coming from the Bayesian approach.

## Estimating the fraction of homotypic HVC_{(RA)} synapses

In order to estimate the homotypic fraction of all excitatory synapses onto $HVC_{(RA)}$ cells, we determined the density of homotypic synapses by counting the number of double labeled synapses and correcting it for the axonal labeling efficiency. Labeling efficiency was estimated by comparing the volume density of labeled axon length by inspecting 300 randomly placed 1 $\mu m^3$ cubes with the density expected for $HVC_{(RA)}$ neurons using published estimates for their total number (*Wang et al., 2002*) and the average axonal path length from LM reconstructions. To count the number of double labeled synapses, BDA-labeled dendrites were searched by an expert annotator for synapses with labeled axons by following them in KNOSSOS at the full voxel resolution, instructed to annotate also synapses with weak labeling. The found synapses were then scrutinized by JK and the result was confirmed by ML and SB.

All error estimates were calculated assuming independence of the errors using the variance formula for error propagation.

## Acknowledgements

This research was supported by the National Institutes of Health (Grant R01NS075044) (M.A.L.) and (F31 NS084767) (S.E.B.), the New York Stem Cell Foundation (M.A.L.), the Max Planck Society (J.K., F.S. R.T.N., M.O., W.D), the Bernstein Center for Computational Neuroscience, Boehringer Ingelheim Foundation (J.K., F.S.), the German Federal Ministry of Education and Research Grant BMBF/FKZ 01GQ1002 (M.O.), and the European Research Council (M.O.) under the European Union's Horizon 2020 research and innovation program (grant agreement No. 633428). We thank L. Gibb, M. Halassa, D. Jin, K. Katlowitz, G. Kosche, M. Picardo A. Reyes, D. Rinberg, R. Tremblay, and D. Vallentin for comments on the manuscript. We are grateful to A. Al-Shaboti, A. Andrade Garcia, E. Atsiatorme, D. Baltissen, S. Bassler, F. Bassler, J. Bessler, D. Dimitrov, M. Ederer, T. Eliguezel, O. Fedorashko, L. Fey, J. Foehr, A. Gaubatz, S. Gottwalt, M. Gross, J. Hammerich, L. Hammes, H. Hees, J. Hendricks, J. Huether, S. Hutzl, R. Janssen, K. Kappler, F. Kaufhold, K. Kehrle, K. Kiesl, J. Kupke, J. Loeffler, S. Mertens, O. Mueller, T. Noel, L. Obenauer, G. Patzer, J. Pellegrino, D. Raica, C. Sandhof, M. Schramm, J. Schwab, D. Schwarz, J. Serwani, A. Sons, J. Tytko, D. Wachmann, and S. Zaschke for help with EM data analysis, J. Kuhl for help with figure preparation, and J. Tritthardt for building custom electronics.

## Additional information

### Funding

| Funder | Grant reference number | Author |
|---|---|---|
| Max Planck Society | Max Planck | Jörgen Kornfeld<br>Rajeevan T Narayanan<br>Fabian Svara<br>Marcel Oberlaender<br>Winfried Denk |
| Boehringer Ingelheim Fonds | | Jörgen Kornfeld<br>Fabian Svara |
| National Institutes of Health | F31 NS084767 | Sam E Benezra |
| Bernstein Center for Computational Neuroscience Tübingen | | Rajeevan T Narayanan<br>Marcel Oberlaender |
| European Research Council | 633428 | Rajeevan T Narayanan<br>Marcel Oberlaender |
| German Federal Ministry of Education and Research Grant | | Rajeevan T Narayanan<br>Marcel Oberlaender |
| European Union's Horizon 2020 | | Rajeevan T Narayanan<br>Marcel Oberlaender |
| National Institutes of Health | R01NS075044 | Michael A Long |
| New York Stem Cell Foundation | NYSCF-R-NI18 | Michael A Long |
| Rita Allen Foundation | Rita Allen | Michael A Long |

The funders had no role in study design, data collection and interpretation, or the decision to submit the work for publication.

### Author contributions

JK, Conceptualization, Data curation, Software, Formal analysis, Funding acquisition, Validation, Investigation, Visualization, Methodology, Writing—original draft, Writing—review and editing; SEB, Conceptualization, Data curation, Formal analysis, Validation, Investigation, Visualization, Methodology, Writing—original draft, Writing—review and editing; RTN, Data curation, Investigation,

Methodology; FS, RE, Software, Methodology; MO, Resources, Supervision, Funding acquisition, Validation, Project administration, Writing—review and editing; WD, Conceptualization, Resources, Data curation, Formal analysis, Supervision, Funding acquisition, Validation, Investigation, Visualization, Methodology, Writing—original draft, Project administration, Writing—review and editing; MAL, Conceptualization, Resources, Data curation, Software, Formal analysis, Supervision, Funding acquisition, Validation, Investigation, Visualization, Methodology, Writing—original draft, Project administration, Writing—review and editing

### Author ORCIDs
Jörgen Kornfeld, http://orcid.org/0000-0002-2547-8700
Sam E Benezra, http://orcid.org/0000-0001-7889-898X
Robert Egger, http://orcid.org/0000-0003-0849-6904
Winfried Denk, http://orcid.org/0000-0002-0704-6998
Michael A Long, http://orcid.org/0000-0002-9283-3741

### Ethics

Animal experimentation: This study was performed in strict accordance with the recommendations in the Guide for the Care and Use of Laboratory Animals of the National Institutes of Health. All of the animals were handled according to approved institutional animal care and use committee (IACUC) protocols of the New York University Medical Center. Our songbird protocol, entitled 'Understanding birdsong circuitry', was recently renewed. The protocol number is 161102-01.

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
