## [Decision Letter]

Thank you for submitting your article "EM connectomics reveals axonal target variation in a sequence generating network" for consideration by *eLife*. Your article has been favorably evaluated by a Senior Editor and three reviewers, one of whom, Karel Svoboda (Reviewer #1), is a member of our Board of Reviewing Editors.

The reviewers have discussed the reviews with one another and the Reviewing Editor has drafted this decision to help you prepare a revised submission.

This is an interesting paper on the detailed connectivity in HVC, a nucleus in the zebrafish brain that is critical for production of stereotyped song. A subclass of HVC neurons (those that project to RA) produce precise sequences (sequential neural activity patterns) of activity. However, the circuit and network properties underlying sequential activity patterns are not understood. Computational models have focused on local excitation to produce this activity. However, excitatory connections between RA-projecting HVC neurons have been difficult to find in brain slice recordings, arguing against the models. Kornfeld et al. show that proximal axons of RA-projecting HVC neurons synapse onto inhibitory interneurons; in contrast, more distal axonal segments synapse onto excitatory neurons, consistent with chain models.

This is the first serial EM reconstruction study of songbird brain, and therefore this manuscript could be an important resource for songbird researchers. The paper contains a wealth of anatomical data that will be useful for unraveling the circuit mechanisms underlying sequence generation in this circuit.

Please revise the paper in response to the points below. Three major points need addressing:

1) Data resource.

2) Conceptual issues. Mainly, the reviewers feel that this report has little to say about chain models and alternatives.

3) Technical issues. Mainly, the classification of symmetric and asymmetric synapses is not well supported.

Data resource issues:

The underlying EM data should be shared upon publication. This would be consistent with recent trends in sharing of volume EM data accompanying publication of biological results (Kasthuri et al., 2015; Bock et al., 2011; Takemura et al., 2013; https://www.janelia.org/project-team/flyem/data-and-software-release; and many others at http://www.openconnectomeproject.org/). Traced skeletons should also be provided.

Approach: the authors confirmed the morphology of HVC_(RA)_ cells by using single-cell labeling and light microscopy (LM). Most (95%) of the dendrites of HVC_(RA)_ cells are sufficiently compact to be covered by the image size of the serial block-face electron microscopy (SBEM) (166 x 166 x 77 μm^3^).

From a postsynaptic perspective, SBEM revealed that the synapses onto an HVC_(RA)_ cell are dominated by inhibitory (symmetric) synapses rather than excitatory (asymmetric) synapses (60%: 40%, main text, sixth paragraph). This fraction is surprising because it is different from results described in the mammalian literature (but see point 1 below). After taking into account the ratio of BDA-labelled (retrogradely labelled) HVC_(RA)_ cells relative to the total number of HVC_(RA)_ cells, HVC_(RA)_ – HVC_(RA)_ connections are estimated to be ~15% of all the excitatory inputs (main text, eighth paragraph) to a single HVC_(RA)_ cell.

From a presynaptic perspective, the authors also found that the probability of making a connection to another HVC_(RA)_ neuron depends on the distance between the presynaptic axon and its cell body of origin. At a short distance (<100μm) from the cell body, most (95%) of the synapses that an HVC_(RA)_ axon makes are on inhibitory neurons, whereas few (3%) are made on other HVC_(RA)_ cells. At greater distances from the soma, the axon tends to make more synapses on other HVC_(RA)_ cells (36.7% of its presynaptic axon terminals, main text, eleventh paragraph). On average, the authors estimated that about a quarter (24%) of all incoming excitatory synapses are homotypic (HVC_(RA)_ -HVC_(RA)_ synapses. The discrepancy between this number and the estimate from post-synaptic side could be due to the difficulty of defining the presynaptic terminal compared to postsynaptic structure.

Although it was already known that HVC_(RA)_ cells make connection to other HVC_(RA)_ cells, the distance dependent connectivity is a novel finding. This is of interest to those who are interested in the network architecture of HVC.

The core analysis for the distance dependent connectivity is that orphaned axons are further from their parent somata and make a higher proportion of excitatory synapses. This is a one bit analysis, which is ok. Beyond that, the attempts at quantitative analysis using Baysian approaches is not convincing.

Major conceptual issue 1:

Although the HVC_(RA)_ – HVC_(RA)_ interaction is the major focus of this paper, the results here show that only a quarter of excitatory inputs onto an HVC_(RA)_ cell are from other HVC_(RA)_ cells. The majority of excitatory inputs to HVC_(RA)_ cells remain to be described. Can the authors provide more information about the remaining ~75% of excitatory inputs? For example, major afferents of HVC are NIf and Uva. In addition to HVC_(X)_ cell type, it is reasonable to assume they (NIf, Uva, HVC_(X)_) would comprise the remaining excitatory inputs to HVC_(RA)_ cell. Since some of these other inputs are likely to be important for singing (especially those from Uva), it seems important to know their identity. As it stands, this paper failed to provide the majority of input to HVC_(RA)_.

Major conceptual issue 2:

These results do not serve to support a chain model. The authors acknowledge this limitation (main text, fifteenth paragraph), but it seems that much of the significance of this study rests on knowing whether the interconnected HVC_(RA)_ neurons are actually parts of functional chains that display sequential activity, as introduced in this manuscript. Testing this idea would require a combination of functional imaging and SBEM methods. This is a high bar. But as it stands, the interconnected neurons could be a cluster of somewhat distributed cells that fire together.

Out another way. The network structure presented here may be consistent with, but not indicative of, a role in sequence generation. This network structure is also consistent with winner-take-all dynamics and with synchrony detection. Is there an argument the authors can make that this network structure is specifically supportive of a role in sequence generation?

Re: "Both observations support the notion that the neural sequences that pace the song are generated by global synaptic chains embedded within local inhibitory networks."

"Are consistent with the notion" would be more accurate. (see also main text, sixteenth paragraph).

Major technical issues

1) Identification of symmetric vs. asymmetric synapses is questionable. The data and methods presented are insufficient. Figure 1 is too small; Figure 1—figure supplement 1 shows only one reasonably clear asymmetric contact (panel D). Since there is no post-staining with this imaging method, post-synaptic densities are unamplified by a mordant. Misclassification biased toward symmetric contacts is therefore possible. Since this is a potential bias in the underlying data, it would not be detected by the verification method, i.e. to have a second expert independently classify synapses to obtain consistent results. Misclassification would be consistent with the surprising finding that these dendrites have unusually high fraction of inhibitory input (~60%). Although it is likely the authors have developed expertise to classify these synapses, despite the above issues, the data presented in the paper are not sufficient to convince the reader. At a minimum, more raw data and examples of classification will need to be shown. If this does not suffice, an independent verification of classification correctness should be devised. For example, axons could be traced from identified excitatory and inhibitory somata in the volume, and the synapses these axons make could be randomly presented for classification to a blinded party. If classification in this data set is correct and unbiased, all synapses arising from excitatory axons should be classified as asymmetric, and all axons arising from inhibitory neurons should be symmetric etc.

2) Re: "Although ~70% (174 out of 248) of dendritic branches reached the boundary of the EM data set and were thus incomplete, many reconstructions included some of the most distal inputs (median ± SD of maximum soma distances: 90.9 ± 8.6 μm and 116.7 ± 29.4 for EM and LM, respectively) and should therefore sample the full gamut of input types."

This assertion is poorly motivated. The LM data had a mean dendritic path length of 3187 μm (main text, fifth paragraph), whereas the EM data had a maximum path length of 1956 μm. The maximum intact dendritic path length discovered in the EM volume therefore failed to reach the observed mean dendritic path length observed at the light level. The authors find that the distribution of synapses changes proximally; what data are available to suggest that this is not also the case distally? If none, it would be better to acknowledge this limitation, and leave the question open for future study.

---

## [Author Response]

*[…] Please revise the paper in response to the points below. Three major points need addressing:*

*1) Data resource.*

*2) Conceptual issues. Mainly, the reviewers feel that this report has little to say about chain models and alternatives.*

*3) Technical issues. Mainly, the classification of symmetric and asymmetric synapses is not well supported.*

*Data resource issues:*

*The underlying EM data should be shared upon publication. This would be consistent with recent trends in sharing of volume EM data accompanying publication of biological results (Kasthuri et al., 2015; Bock et al., 2011; Takemura et al., 2013; https://www.janelia.org/project-team/flyem/data-and-software-release; and many others at http://www.openconnectomeproject.org/). Traced skeletons should also be provided.*

We will release all raw data as well as traced skeletons to the public upon publication, and we now provide a URL within the main body of the paper to the online resources containing these materials. We are also sharing all EM analysis code, which can be found under the same URL.

*Approach: the authors confirmed the morphology of HVC_(RA)_ cells by using single-cell labeling and light microscopy (LM). Most (95%) of the dendrites of HVC_(RA)_ cells are sufficiently compact to be covered by the image size of the serial block-face electron microscopy (SBEM) (166 x 166 x 77 μm^[3]^).*

*From a postsynaptic perspective, SBEM revealed that the synapses onto an HVC_(RA)_ cell are dominated by inhibitory (symmetric) synapses rather than excitatory (asymmetric) synapses (60%: 40%, main text, sixth paragraph). This fraction is surprising because it is different from results described in the mammalian literature (but see point 1 below). After taking into account the ratio of BDA-labelled (retrogradely labelled) HVC_(RA)_ cells relative to the total number of HVC_(RA)_ cells, HVC_(RA)_ – HVC_(RA)_ connections are estimated to be ~15% of all the excitatory inputs (main text, eighth paragraph) to a single HVC_(RA)_ cell.*

*From a presynaptic perspective, the authors also found that the probability of making a connection to another HVC_(RA)_ neuron depends on the distance between the presynaptic axon and its cell body of origin. At a short distance (<100μm) from the cell body, most (95%) of the synapses that an HVC_(RA)_ axon makes are on inhibitory neurons, whereas few (3%) are made on other HVC_(RA)_ cells. At greater distances from the soma, the axon tends to make more synapses on other HVC_(RA)_ cells (36.7% of its presynaptic axon terminals, main text, eleventh paragraph). On average, the authors estimated that about a quarter (24%) of all incoming excitatory synapses are homotypic (HVC_(RA)_ -HVC_(RA)_ synapses. The discrepancy between this number and the estimate from post-synaptic side could be due to the difficulty of defining the presynaptic terminal compared to postsynaptic structure.*

*Although it was already known that HVC_(RA)_ cells make connection to other HVC_(RA)_ cells, the distance dependent connectivity is a novel finding. This is of interest to those who are interested in the network architecture of HVC.*

Importantly, the reviewers point out that connectivity between HVC_(RA)_ neurons had been already demonstrated, but we should certainly emphasize that the approaches used to test this connectivity (paired intracellular recordings in slices) suggested that it was exceptionally weak. For instance, in an earlier study, only 1 out of 141 HVC_(RA)_ pairs were synaptically coupled (Kosche et al., J Neurosci, 2015), which seemed to be inconsistent with the model of a synaptic (or synfire) chain. A major finding of the present study – in addition to the distance dependence – is the fact that each HVC_(RA)_ neuron is forming nearly 200 connections onto other HVC_(RA)_ neurons.

*The core analysis for the distance dependent connectivity is that orphaned axons are further from their parent somata and make a higher proportion of excitatory synapses. This is a one bit analysis, which is ok. Beyond that, the attempts at quantitative analysis using Baysian approaches is not convincing.*

We agree that a more convincing approach would be to analyze connectivity within a much larger volume in which the distances of each synapse from the soma could be directly measured. However, in our study, we developed an innovative method for exploiting the fact that branching patterns in HVC_(RA)_ neurons vary with the distance of a given axonal segment from the cell body. To establish this relationship, we performed the first full reconstructions of the local axonal arbors of HVC_(RA)_ neurons and assembled a data set of 15 such cells in light microscopy. We then used this information to perform two separate and independent analyses (a simulation as well as the Bayesian approach) to estimate the distances of our identified (retrogradely labeled) axonal pieces from the cell bodies. Both gave very similar results. We would be grateful if the reviewer could provide us with a more detailed explanation of why he/she doesn’t find our Bayesian approach convincing.

*Major conceptual issue 1:*

*Although the HVC_(RA)_ – HVC_(RA)_ interaction is the major focus of this paper, the results here show that only a quarter of excitatory inputs onto an HVC_(RA)_ cell are from other HVC_(RA)_ cells. The majority of excitatory inputs to HVC_(RA)_ cells remain to be described. Can the authors provide more information about the remaining ~75% of excitatory inputs? For example, major afferents of HVC are NIf and Uva. In addition to HVC_(X)_ cell type, it is reasonable to assume they (NIf, Uva, HVC_(X)_) would comprise the remaining excitatory inputs to HVC_(RA)_ cell. Since some of these other inputs are likely to be important for singing (especially those from Uva), it seems important to know their identity. As it stands, this paper failed to provide the majority of input to HVC_(RA)_.*

As the reviewers correctly point out, nearly 75% of the excitatory synapses onto HVC_(RA)_ neurons remain to be identified. However, the focus of this study was the connectivity within HVC in light of a whole range of experiments over the years that have suggested that at least the timing within each syllable is generated inside HVC. While studies will undoubtedly attempt to quantify the relative number of connections from other regions (e.g., Uva, NIf, etc.) at the level of single HVC_(RA)_ neurons, many of these connections, such as auditory afferents (Vallentin and Long, 2015), collaterals from HVC_(X)_ neurons (Scharff et al., 2000), and descending fibers from NIf (Cardin et al., 2005) are unlikely to play a role in motor patterning. We now include these points in the Discussion.

*Major conceptual issue 2: These results do not serve to support a chain model. The authors acknowledge this limitation (main text, fifteenth paragraph), but it seems that much of the significance of this study rests on knowing whether the interconnected HVC(RA) neurons are actually parts of functional chains that display sequential activity, as introduced in this manuscript. Testing this idea would require a combination of functional imaging and SBEM methods. This is a high bar. But as it stands, the interconnected neurons could be a cluster of somewhat distributed cells that fire together.*

We agree that our results do not exclusively support a chain model, but they do remove doubts about whether the amount of direct connectivity between HVC_(RA)_ cells is sufficient, in principle, to support a direct synaptic chain, which had be a matter of quite some debate as a result mostly of paired electrical recordings. Our finding that excitatory connections are substantially less frequent from the proximal part of the axon provides an explanation for the results of paired electrical recording experiments and suggests a network architecture that combines multiple neural circuit motifs, the lateral inhibition-based winner-take-all circuit and the spatially dispersed chain. It would, of course, be desirable to have a wiring diagram of a large part of HVC together with calcium recordings from a sufficient number of cell to ensure that pairs of cells with closely spaced burst times can be found among them. This is particularly challenging, given that there seems to be little correlation between time and space in HVC. In fact, our findings suggest that cells that fire in succession may be even less likely to be near each other than chance would suggest. We now discuss this problem in the manuscript but the combined activity/connectivity experiment is certainly beyond the scope of this work.

*Out another way. The network structure presented here may be consistent with, but not indicative of, a role in sequence generation. This network structure is also consistent with winner-take-all dynamics and with synchrony detection. Is there an argument the authors can make that this network structure is specifically supportive of a role in sequence generation?*

We believe that a feed-forward excitatory synaptic chain is likely to be an important circuit motif for sequence generation, however, this is not to say that other mechanisms are *not* involved in this process. For instance, in previous network models, we have shown the importance of synchrony detection (Long et al., 2010), and others have proposed that stable sequences can be established through winner-take-all networks (Mostafa and Indiveri, 2014).

*Re: "Both observations support the notion that the neural sequences that pace the song are generated by global synaptic chains embedded within local inhibitory networks."*

*"Are consistent with the notion" would be more accurate. (see also main text, sixteenth paragraph).*

Done.

*Major technical issues*

*1) Identification of symmetric vs. asymmetric synapses is questionable. The data and methods presented are insufficient. Figure 1 is too small; Figure 1—figure supplement 1 shows only one reasonably clear asymmetric contact (panel D). Since there is no post-staining with this imaging method, post-synaptic densities are unamplified by a mordant. Misclassification biased toward symmetric contacts is therefore possible. Since this is a potential bias in the underlying data, it would not be detected by the verification method, i.e. to have a second expert independently classify synapses to obtain consistent results. Misclassification would be consistent with the surprising finding that these dendrites have unusually high fraction of inhibitory input (~60%). Although it is likely the authors have developed expertise to classify these synapses, despite the above issues, the data presented in the paper are not sufficient to convince the reader. At a minimum, more raw data and examples of classification will need to be shown. If this does not suffice, an independent verification of classification correctness should be devised. For example, axons could be traced from identified excitatory and inhibitory somata in the volume, and the synapses these axons make could be randomly presented for classification to a blinded party. If classification in this data set is correct and unbiased, all synapses arising from excitatory axons should be classified as asymmetric, and all axons arising from inhibitory neurons should be symmetric etc.*

Our findings strongly depend on the proper identification symmetric and asymmetric synapses. To address the concerns raised by reviewers, we have included significant new data and analysis. We have now analyzed 50 additional synapses. In short, the reviewers propose an independent verification of the synapse type classification based on cells identified through other means (i.e. somatic and dendritic appearance). We performed this verification of the synapse type classification by randomly sampling 31 synapses from the axons of identified interneurons and 19 synapses from the axons of one HVC_(RA)_ neuron (8 synapses) and two HVC_(X)_ neurons (11 synapses), and we presented these synapses in a randomized and blind manner to the expert annotator who performed the original classification. All 50 synapses were correctly assigned, confirming our classification approach. We now include these data within the main body of the text.

*2) Re: "Although ~70% (174 out of 248) of dendritic branches reached the boundary of the EM data set and were thus incomplete, many reconstructions included some of the most distal inputs (median ± SD of maximum soma distances: 90.9 ± 8.6 μm and 116.7 ± 29.4 for EM and LM, respectively) and should therefore sample the full gamut of input types."*

*This assertion is poorly motivated. The LM data had a mean dendritic path length of 3187 μm (main text, fifth paragraph), whereas the EM data had a maximum path length of 1956 um. The maximum intact dendritic path length discovered in the EM volume therefore failed to reach the observed mean dendritic path length observed at the light level. The authors find that the distribution of synapses changes proximally; what data are available to suggest that this is not also the case distally? If none, it would be better to acknowledge this limitation, and leave the question open for future study.*

Our plot in Figure 1 indicates a flat distribution of the synapse densities for asymmetric and symmetric synapses for soma distances ranging from 50 µm to 120 µm. Our LM experiments indicate that 95% of the dendritic path length is contained within 100 µm around the soma, which suggests that we would have detected an input bias if one existed. However, given the fact that a significant amount of the EM path length was excluded from our volume, we now state this limitation within the text.